# Nanoemulsions of *Jasminum humile* L. and *Jasminum grandiflorum* L. Essential Oils: An Approach to Enhance Their Cytotoxic and Antiviral Effects

**DOI:** 10.3390/molecules27113639

**Published:** 2022-06-06

**Authors:** Khaled Ahmed Mansour, Mona El-Neketi, Mohamed-Farid Lahloub, Ahmed Elbermawi

**Affiliations:** 1Department of Pharmacognosy, Faculty of Pharmacy, Mansoura University, Mansoura 35516, Egypt; khali@horus.edu.eg (K.A.M.); mfil@mans.edu.eg (M.-F.L.); 2Department of Pharmacognosy, Faculty of Pharmacy, Horus University in Egypt, New Damietta 34517, Egypt

**Keywords:** *Jasminum*, humile, *grandiflorum*, nanoemulsion, HAV, HSV-1, HepG-2, MCF-7, THP-1

## Abstract

Unprecedented nanoemulsion formulations (NE) of *Jasminum humile* and *Jasminum grandiflorum* essential oils (EO) were prepared, and examined for their cytotoxic and antiviral activities. NE characterization and stability examination tests were performed to ensure formula stability. The antiviral activity was determined against hepatitis A (HAV) and herpes simplex type-1 (HSV-1) viruses using MTT assay, while the cytotoxic potential was determined against liver (HepG-2), breast (MCF-7), leukemia (THP-1) cancer cell lines and normal Vero cells. Statistical significance was determined in comparison with doxorubicin as cytotoxic and acyclovir as antiviral standard drugs. GC-MS analysis indicated twenty four compounds in the EO of *J. humile* and seventeen compounds in the EO of *J. grandiflorum*. Biological investigations of pure EOs revealed weak cytotoxic and antiviral effects. Nevertheless, their NE formulations exhibited high biological value as cytotoxic and antiviral agents. NE formulations also showed feasible selectivity index for the viral-infected and cancer cells (especially HepG-2) than normal Vero cells. Both nanoemulsions showed lower IC_50_ than standard doxorubicin against HepG-2 (26.65 and 22.58 vs. 33.96 μg/mL) and MCF-7 (36.09 and 36.19 vs. 52.73 μg/mL), respectively. The study results showed the dramatic effect of nanoemulsion preparation on the biological activity of EOs and other liposoluble phytopharmaceuticals.

## 1. Introduction

*Jasminum* is a genus of flowering plants (approximately 600 species), belonging to family Oleaceae, usually used in perfume industries and ornamental purposes due to their bright colored flowers and characteristic fragrances. Traditionally, the essential oils (EOs) of different *Jasminum* species were used in aromatherapy for the treatment of diarrhea, fever, abdominal spasms, conjunctivitis, skin inflammations, bronchial asthma and uterine hemorrhage [1]. Nowadays, several reports investigated the biological activities of the EOs of several *Jasminum* species. *Jaminum sambac* EO was reported to exhibit antibacterial activity and to suppress puerperal lactation [2] in addition to its antidepressant and mood uplifting properties [3]. *Jasminum officinale* EO was reported to have antiviral activity against hepatitis B virus [4]. Only few studies investigated the biological activities of *Jasminum grandiflorum* EO, much less *Jasminum humile* EO. The EO of *Jasminum grandiflorum* was examined for its antimicrobial activity against Gram-positive and Gram-negative bacteria as well as candidiasis [5]. *Jasminum grandiflorum* was reported to have cytotoxic activity against brain cancer cell line [6].

In recent times, there has been more interest in the nanoscience and nanotechnologies applications. Nanotechnology modifies and develops the properties of drugs by converting them into their nanoparticles. It has numerous applications including drug delivery and disease diagnostics [7]. The small molecule-based nanotechnology may reduce the side effects, enhance the potency, and deliver drugs in a well-targeted manner through increasing their permeability and retention effects [8]. As droplets decrease in size, reaching nano-diameter; the biological activity of lipophilic constituents prepared as nanoemulsions increases. This is due to the higher surface area obtained, and the easier transport of the effective drugs through bio-membranes [9]. Although several previous studies were reported about the biological activities of the EOs of different *Jasminum* species, there are no reports, as far as we know, investigating the potency of their nanoemulsion preparations on the biological effects.

In 2020, breast cancer was found to be first in line among cancer diagnosed cases. Liver cancer was one of the top three, in the list of cancer-related deaths [10]. Both cancer types are pervasive in Egypt, ranking as first and second leading causes of cancer cases [11], while leukemia was responsible for 35.6% of diagnosed cancer cases among children [12]. In this study, chemical profiling was performed for the EOs of two oleaceous plants; *Jasminum humile* L. and *Jasminum grandiflorum* L. using GC-MS analysis technique. Moreover, nanoemulsion formulations were prepared for both EOs (JhEO and JgEO), then the pure EOs and NE formula were examined biologically for their cytotoxic and antiviral effects to investigate the possible activity enhancement by nanoemulstion preparation. The cytotoxic effects were examined against HepG-2, MCF-7 and THP-1 human cancer cell lines in comparison with doxorubicin as a standard. Statistical significance was determined and normal Vero cells were used to evaluate their selectivity towards cancer cells. The antiviral activities were also investigated using MTT assay against hepatitis A virus (HAV) and herpes simplex virus type 1 (HSV-1) and results were compared to the standard drug acyclovir.

## 2. Materials and Methods

### 2.1. Plant Materials

The fresh flowers of *Jasminum humile* L. and *Jasminum grandiflorum* L. (250 gm, each) were collected in the early morning during May 2021 from the botanical garden of Kafr EL-Sheikh university, Egypt and were attested by Dr. Ibrahim Mashaly, Professor of Ecology, Faculty of Science, Mansoura University, Egypt. The essential oils (EOs) were extracted by hydro-distillation for almost 8 h; according to the European pharmacopoeia (European Pharmacopoeia, 1975) using Clevenger’s cohobation apparatus to produce 2 mL and 2.5 mL, respectively of yellow colored EOs. The collected oils were dehydrated over anhydrous sodium sulfate and then they were stored in sealed vials at low temperature for further (GC/MS) analysis and biological investigations.

### 2.2. Capillary Gas Chromatography-Mass Spectrometry (GC/MS) Analysis

The GC/MS analysis was carried out at the faculty of postgraduate studies for advanced sciences (the central laboratory), Beni Suef. Egypt. Identification of the components was confirmed by comparing their mass spectral fragmentation patterns and retention indices with the previously reported literature [13,14,15,16,17] as well as the mass spectral NIST/ChemStation database.

### 2.3. Preparation of Nanoemulsion Formulations

JhEO-NE and JgEO-NE formula were prepared by mixing 1% *w*/*w* of each EO with 8% *w*/*w* glycerol monoacetate (GMA). Tween 80 (surfactant) was mixed with labrosol (co-surfactant) in an equal ratio, and the mixture (30% *w*/*w*) was added to EO-GMA. Finally water (61% *w*/*w*) was dropped in order to obtain an apparent and clear NE [18]. The prepared JhEO-NE and JgEO-NE formula were subjected to thermodynamic stability studies and self-nanoemulsification efficiency tests according to the International Conference for Harmonization (ICH) guidelines. Freshly prepared formula were packaged in glass bottles and subjected to different storage conditions of refrigeration at (5 ± 3 °C) and different ambient conditions over a period of 3 months. Physical assessment of the samples was achieved by visual inspection of phase separation, color and/or odor change and pH measurements at zero time (freshly prepared formula), as well as after storage periods of 1, 2 and 3 months. The particle size, polydispersity index (PDI) and zeta potential (ZP) were also measured (using Malvern Zetasizer Nanoseries, Malvern Instruments Limited, UK).

### 2.4. Determination of Sample Cytotoxicity on Cancer Cells

Cytotoxic potential was measured against normal Vero cells (ATCC; CCL81-22), as well as HepG-2 (ATCC; HB-8065), MCF-7 (ATCC; HTB-22), and THP-1 (ATCC; TIB-202) human cancer cell lines, obtained from American Type Culture Collection (Rockville, MD, USA), using a microplate 3-(4,5-dimethythiazole-2yl)-2,5-diphenyl-tetrazolium bromide (MTT) method. Experiments were repeated for three times with doxorubicin (Sigma, USA) as the positive control and 0.1% DMSO media as the negative control. Cells were grown in DMEM supplemented with 10% FBS, 100 µg/mL of streptomycin, 100 units/mL of penicillin, 0.07% NaHCO_3_ and 2 mM L-glutamine and maintained at 37 °C in humidified 5% CO_2_ atmosphere. The cultures were maintained at 37 °C in a humidified 5% CO_2_ atmosphere.

### 2.5. Determination of Antiviral Activity

Hepatitis A and Herpes simplex-1 viruses were provided by Dr. Mohammed Ali, Virology Lab., Faculty of Medicine, Al-Azhar University, Egypt. Cells were grown in DMEM supplemented with 10% FBS, 100 µg/mL of streptomycin, 100 units/mL of penicillin, 0.07% NaHCO_3_ and 2 mM L-glutamine and maintained at 37 °C in humidified 5% CO_2_ atmosphere. The antiviral assay was performed using a microplate 3-(4,5-dimethythiazole-2yl)-2,5-diphenyl-tetrazolium bromide (MTT) method [19,20]. Experiments were repeated for three times with acyclovir (Sigma-Aldrich, St. Louis, MO, USA) as the positive control and 0.1% DMSO media as the negative control.

### 2.6. MTT Protocol

A 96 well tissue culture plate was inoculated with 1 × 10^5^ cells/ml (100 µL/well) and incubated at 37 °C for 24 h to develop a complete monolayer sheet. Growth medium was decanted from 96 well micro titer plates after confluent sheet of cells were formed, cell monolayer was washed twice with wash media. Two-fold dilutions of tested samples were made in RPMI medium with 2% serum (maintenance medium). 0.1 mL of each dilution was tested in different wells leaving 3 wells as control, receiving only maintenance medium. The plate was incubated at 37 °C and examined. Cells were checked for any physical signs of toxicity e.g., partial or complete loss of the monolayer, rounding, shrinkage, or cell granulation. The cytotoxic potential was investigated utilizing doses of (1000, 500, 250, 125, 62.5, 31.25, and 15.62 µg/mL), while the antiviral activity was assessed by applying (31.25, 15.62, and 7.81 µg/mL) of each sample.

MTT solution was prepared (5 mg/mL in PBS) (Bio Basic Inc.; Markham, Ontario, Canada) and 20 µL MTT solution were added to each well. A shaking table was used (150 rpm for 5 min) to thoroughly mix the MTT into the media and incubated (37 °C, 5% CO_2_) for 1–5 h to allow the MTT to be metabolized. Media was dumped off and the plate was dried on paper towels—if necessary—to remove residues. Formazan (MTT metabolic product) was resuspended in 200 µL DMSO, then placed on a shaking table −150 rpm for 5 min—to thoroughly mix the formazan into the solvent. Optical density was measured at 560 nm (and subtract background at 620 nm), which is directly correlated with cell quantity.

### 2.7. Calculation of the Selectivity Index (SI)

Selectivity indices (SI) were obtained after dividing CC_50_ (the half maximal inhibitory concentration of normal Vero cells) by the specific IC_50_ of cancer cells and viral infected cells. Selectivity indices are used to evaluate the cytotoxic potential and antiviral activity relative to the normal cells toxicity; where high (SI) indicates high potency and low cellular toxicity [21,22,23].

### 2.8. Statistical Analysis

All statistical analyses were performed using GraphPad Prism version 9.2.0 to calculate the half-maximal inhibitory concentration (IC_50_) and the half-maximal cytotoxic concentration (CC_50_) where the level of significance was set at (*p* > 0.05). Quantitative data were expressed as mean ± standard deviation (SD). GraphPad Prism version 9.2.0 was also used to create multiple bar charts of the antiviral activity and cell viability.

## 3. Results and Discussion

### 3.1. GC-MS Analysis of the Essential Oil

Freshly collected flowers of *J. grandiflorum* yielded 0.8% *v*/*w* of a clear, faint yellow, lighter than water, essential oil. The oil components together with the percentage and retention indices are shown in (Table 1). Seventeen compounds (97.7%) were identified in the oil. Esters and sesquiterpenes, are the main constituents, accounting for 49.4% and 25.7%, respectively. Esters are represented by benzyl acetate (32.4%), benzyl benzoate (7.4%), methyl anthranilate (2.5%), benzyl salicylate (2.5%), (3*Z*)-hexenyl benzoate (1.1%), (*Z*)-methyl jasmonate (1.8%) and (*Z*)-methyl epijasmonate (1.6%) as the main constituents, while sesquiterpenes are represented by (*Z*)-nerolidol (11.9%), (*E*, *E*)-*α*-farnesene (7.606%) and (*Z*)-caryophyllene (6.2%) as the main constituents.

Fresh flowers of *J. humile* yielded 1.0% *v*/*w* of a clear, faint yellow, lighter than water, essential oil. The oil components together with the percentage and retention indices are shown in (Table 1). Twenty four compounds (98.0%) were identified in the oil. Esters, monoterpenes and sesquiterpenes are the main constituents, accounting for 27.4%, 26.0% and 26.0%, respectively. Esters are represented by (3*Z*)-hexenyl benzoate (7.429%), benzyl benzoate (6.9%), methyl linoleate (3.1%), benzyl salicylate (2.6%), methyl anthranilate (2.1%), (2*E*, 6*E*)-farnesyl acetate (2.0%), geranyl benzoate (1.7%) and phytol acetate (1.5%) as the main constituents. Monoterpenes are represented by linalool (17.2%), (*Z*)-jasmone (6.6%) and carvone (2.2%) as the main constituents. Sesquiterpenes are represented by (*E*, *E*)-*α*-farnesene (6.9%), (*Z*)-caryophyllene (5.6%), (*Z*)-nerolidol (5.0%), epi-α-cadinol (3.0%), epi-α-muurolol (2.9%) and (2*E*, 6*Z*)-farnesol (2.5%) as the main constituents.

### 3.2. Characterization of Nanoemuslion Formulations (Physical Characterization)

JhEO-NE and JgEO-NE formula passed thermodynamic stability as well as self-nanoemulsification efficiency tests. There was no observed phase separation or physical changes, in color, odor, or pH, over the storage period of 3 months at refrigeration conditions (5 ± 3 °C). They had negative ZP charges of (−18.63 and −229.67), nanometric sizes of (8.03 ± 0.45 and 12.09 ± 0.81 nm) and low PDI values of (0.30 ± 0.04 and 0.27 ± 0.03), respectively.

### 3.3. Cytotoxic Concentrations of Tested Samples on Normal Cells

The half maximal cytotoxic concentration (CC_50_) for the essential oils (EOs) and the nanoemuslion formulatios of *Jasminum grandiflorum* L. (JgEO-NE) and *Jasminum humile* L. (JhEO-NE) was assessed on VERO cells using MTT assay (Table 2 and Table 3). Dose-response curves are shown in (Figure 1). Both EOs (JgEO and JhEO) showed CC_50_ values of 362.83 and 377.33 μg/mL, respectively. Plain-NE showed CC_50_ of 73.80 μg/mL. Nanoemulsion formula showed lower cytotoxicity where JhEO-NE showed CC_50_ of 64.28 μg/mL, while JgEO-NE showed CC_50_ of 62.42 μg/mL.

### 3.4. Cytotoxic Activity of Tested Samples

The cytotoxic activity was evaluated for the two (EOs) as well as their nanoemulsion formulations against three cell lines; HepG-2, MCF-7 and THP-1. The Vero cell line is a non-tumorigenic cell line established from kidney cells of the African green monkey (*Cercopitbecus aetbiops*). It can be used for investigating cell growth, differentiation, and cytotoxicity against non-cancerous cells. This cell line is an excellent in vitro model for investigating cytotoxicity and carcinogenesis due to its well-defined growth pattern and behavior in culture [24,25,26]. Human hepatocellular carcinoma cells (HepG-2) are usually used as an in vitro model for assessing cytotoxicity against liver cancer cells. They preserve several genotypic and phenotypic features of hepatocytes. Their cytotoxicity assay is highly sensitive and specific [27]. MCF-7 is a classic in vitro model usually used for determining the cytotoxicity against breast cancer cells for having several mammary epithelium characteristics, e.g., estradiol processing, and presence of estrogen receptors (ER) in their cytoplasm [28]. THP-1 is a monocytic-derived cell line; commonly used to investigate immune system disorders, and cytotoxicity. Many reports investigated the cytotoxic potential of different drugs on THP-1 leukemia cells; as a model for acute myeloid leukemia [29,30,31,32,33,34,35].

Dose-response curves are shown in (Figure 2, Figure 3 and Figure 4). Both (EOs) showed weak cytotoxic activity against the examined cell lines. JgEO showed IC_50_ value of 324.90 μg/mL against HepG-2, 327.53 μg/mL against MCF-7 and 286.37 μg/mL against THP-1 cells, while JhEO showed IC_50_ value of 289.10 μg/mL against HepG-2, 304.13 μg/mL against MCF-7 and 265.60 μg/mL against THP-1 cells. Yet, the prepared nanoemuslion formulations of the two (EOs) induced significant cytotoxic activity against all cell lines in a concentration-dependent manner (Appendix A), and the results are depicted in (Table 2).

(JhEO-NE) and (JgEO-NE) significantly inhibited cell growth of the liver cancer cell line; HepG-2 at 250, 125, 62.5 and 31.25 μg/mL (compared to the reference drug doxorubicin) (Appendix A). Statistical significance was also observed for both nanoenulsion formulations against the breast cancer cell line; MCF-7 (Appendix A) at all the examined concentrations (1000, 500, 250, 125, 62.5 and 31.25 μg/mL). Only 3 concentrations (1000, 500 and 250 μg/mL) showed statistically significant inhibition of the cell viability of leukemia cell line; THP-1 (Appendix A).

Compared to standard doxorubicin, the nanoformulations (JhEO-NE) and (JgEO-NE) demonstrated lower IC_50_; i.e., higher cytotoxic activity against the two solid tumor cell lines; MCF-7 (showing 36.31 and 36.33 μg/mL, respectively) vs. (53.09 μg/mL for doxorubicin), and HepG-2 (showing 22.58 and 27.08 μg/mL, respectively) vs. (31.83 μg/mL for doxorubicin) (Appendix A). Statistical significance was determined for their IC_50_, indicating high statistical significance against MCF-7 along with HepG-2 cell lines. For leukemia (THP-1) cell line; (JhEO-NE) exhibited an IC_50_ value of (53.57 μg/mL), while (JgEO-NE) showed an IC_50_ value of (56.10 μg/mL) (Appendix A). Plain-NE exhibited an IC_50_ value of 55.85 μg/mL against HepG-2, 54.77 μg/mL against MCF-7 and 85.98 μg/mL against THP-1 cell lines. The cytotoxic potential of both EOs may be associated with (*Z*)-caryophyllene [36], oleic acid [37,38], and linalool; which has been previously reported to have cytotoxic activity against HepG-2 [39], MCF-7 [40], and THP-1 [31].

### 3.5. Antiviral Activity of Tested Samples

The antiviral activity was evaluated at the maximum non-toxic concentration (MNTC) against 100 tissue culture infectious dose TCID50/mL of HAV and HSV-1 viruses. Dose-response curves are shown in (Figure 5 and Figure 6). The examined (EOs) showed very weak antiviral activity against both viruses. JgEO revealed an IC_50_ value of 275.57 μg/mL against HAV and 299.63 μg/mL against HSV-1, while JhEO showed an IC_50_ value of 265.70 μg/mL against HAV and 275.33 μg/mL against HSV-1. However, the prepared nanoemuslion formulations of the two (EOs) induced significant antiviral activity against both viruses in a concentration-dependent manner (Appendix A), and the results are depicted in (Table 3). Statistical significance was determined for their antiviral activities in comparison with the standard drug acyclovir after using various concentrations (Appendix A).

(JhEO-NE) demonstrated a lower IC_50_ i.e., a higher antiviral activity against both viruses. It exhibited an IC_50_ value of 21.80 μg/mL against HAV and 18.49 μg/mL against HSV-1, while (JgEO-NE) showed IC_50_ value of 25.37 μg/mL against HAV and 12.49 μg/mL against HSV-1. Plain-NE was also examined, showing IC_50_ values of 45.24 μg/mL against HAV and 42.11 μg/mL against HSV-1. Several constituents in the chemical composition of both EOs were reported to have antiviral activity, such as linalool [41], nerolidol [42], and straight chain hydrocarbon compounds (acyclic alkanes) e.g., tetracosane [43], and nonacosane [44].

## 4. Conclusions

GC-MS profiling showed the presence of twenty four compounds in the EO of *J. humile* and seventeen compounds in the EO of *J. grandiflorum*. A stable unprecedented nanoemulsion formulation was prepared for each EO. Biological investigation was performed for the pure EOs and NE formulations of both plants, examining their antiviral activity against HAV and HSV-1 viruses, as well as their cytotoxicity against HepG-2, MCF-7 and THP-1 cell lines.

As cytotoxic agents, both NE formula revealed much more activity and selectivity against solid tumor cell lines; HepG-2 and MCF-7 than the liquid tumor THP-1 cells. Their potency was higher and their selectivity index was comparative to that of the standard cytotoxic drug; doxorubicin. A Flow cytometric cell cycle analysis would be recommended; to measure the cells’ distribution in different stages of the cell cycle and evaluate cell proliferation in more detail. On the other hand; although their antiviral activity was enhanced (than the pure EOs), the standard antiviral drug -acyclovir- was more active than both nanoformulations against HAV and HSV-1.

Results showed clearly that NE formulations had much greater potency than pure EOs. Nanoemulsification highly increased the cytotoxic and antiviral activities of the pure EOs of the two plants which provides an excellent way to improve the biological activity of liposoluble natural agents. Moreover, NE formula could be a promising carrier for the drug administration of anticancer agents; which could enhance their efficacy, and reduce both dose and cost. These encouraging in-vitro results also recommend further in-vivo and clinical investigations of the phytopharmaceutical NE formulations as promising approach to enhance the essential oils cytotoxic and antiviral activities.

## Figures and Tables

**Figure 1 molecules-27-03639-f001:**
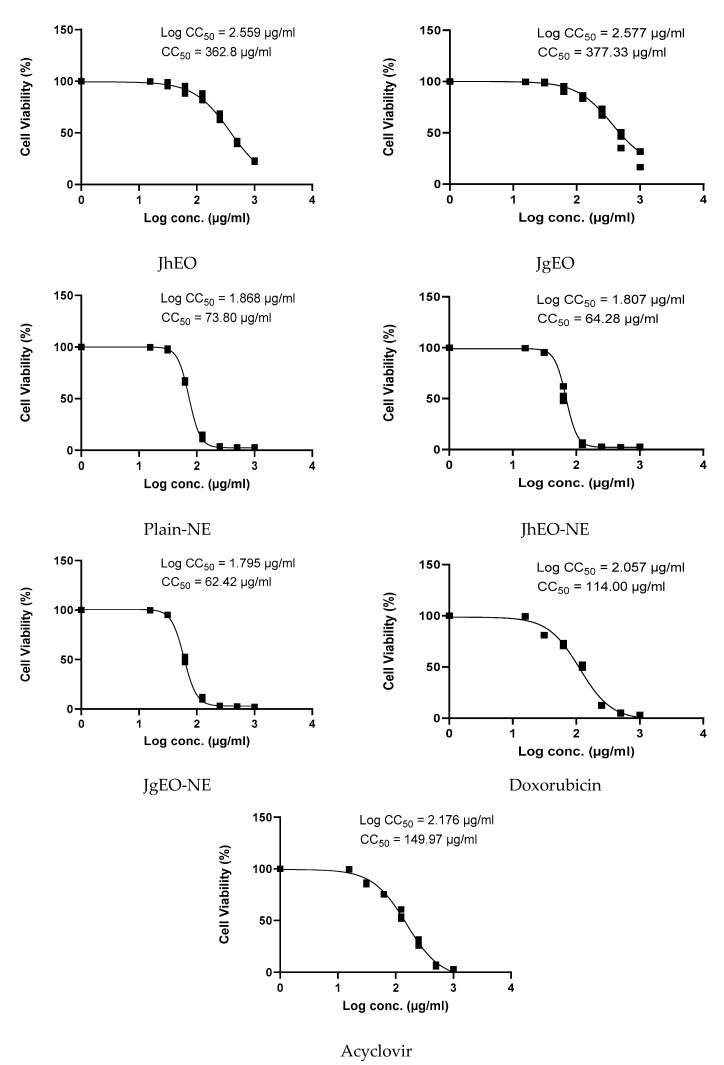
CC_50_ of *J. humile* (JhEO) and *J. grandiflorum* (JgEO) pure essential oils, their nanoemulsion formulations (JhEO-NE) and (JgEO-NE), plain nanoemulsion formulation (Plain-NE) as well as the standard drugs doxorubicin and acyclovir against normal Vero cells.

**Figure 2 molecules-27-03639-f002:**
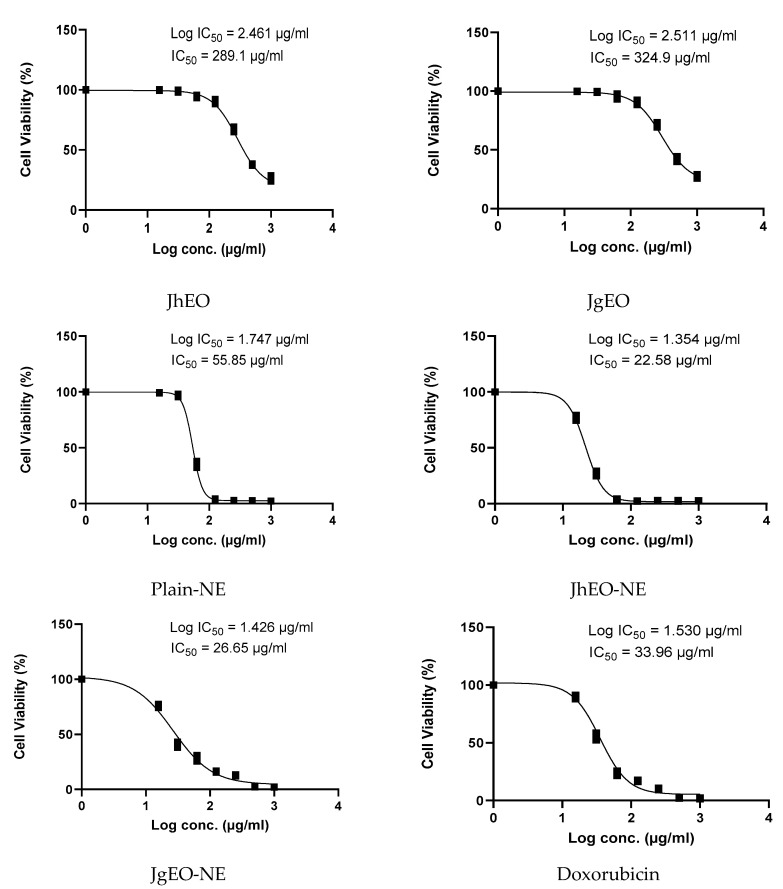
IC_50_ of *J. humile* (JhEO) and *J. grandiflorum* (JgEO) pure essential oils, their nanoemulsion formulations (JhEO-NE) and (JgEO-NE), plain nanoemulsion formulation (Plain-NE) as well as standard doxorubicin against HepG-2 cell line.

**Figure 3 molecules-27-03639-f003:**
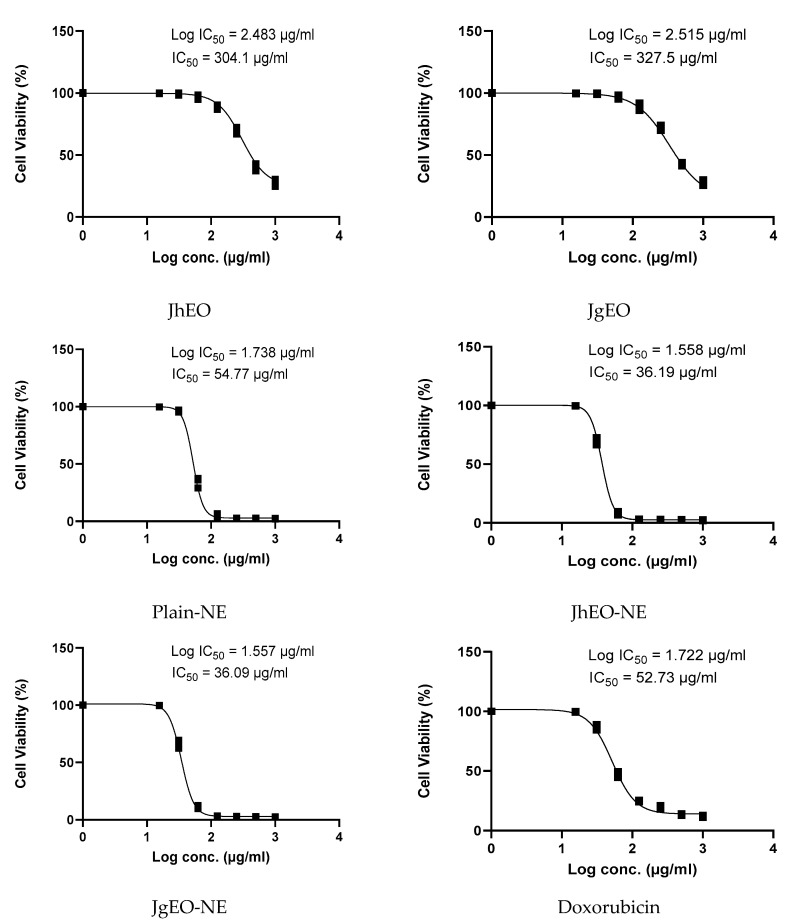
IC_50_ of *J. humile* (JhEO) and *J. grandiflorum* (JgEO) pure essential oils, their nanoemulsion formulations (JhEO-NE) and (JgEO-NE), plain nanoemulsion formulation (Plain-NE) as well as standard doxorubicin against MCF-7 cell line.

**Figure 4 molecules-27-03639-f004:**
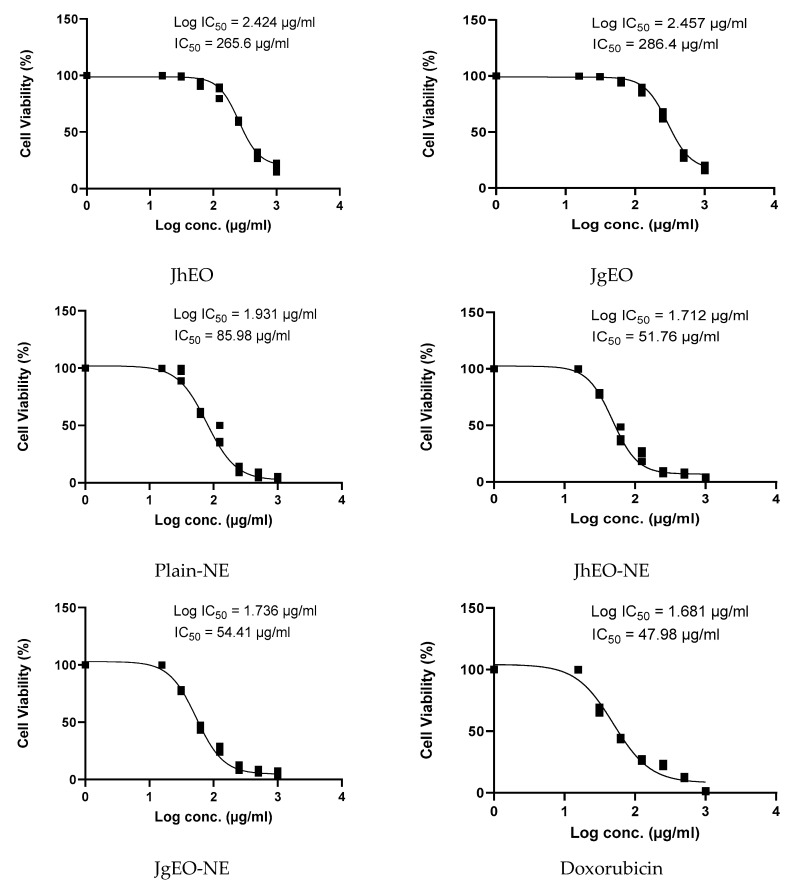
IC_50_ of *J. humile* (JhEO) and *J. grandiflorum* (JgEO) pure essential oils, their nanoemulsion formulations (JhEO-NE) and (JgEO-NE), plain nanoemulsion formulation (Plain-NE) as well as standard doxorubicin against THP-1 cell line.

**Figure 5 molecules-27-03639-f005:**
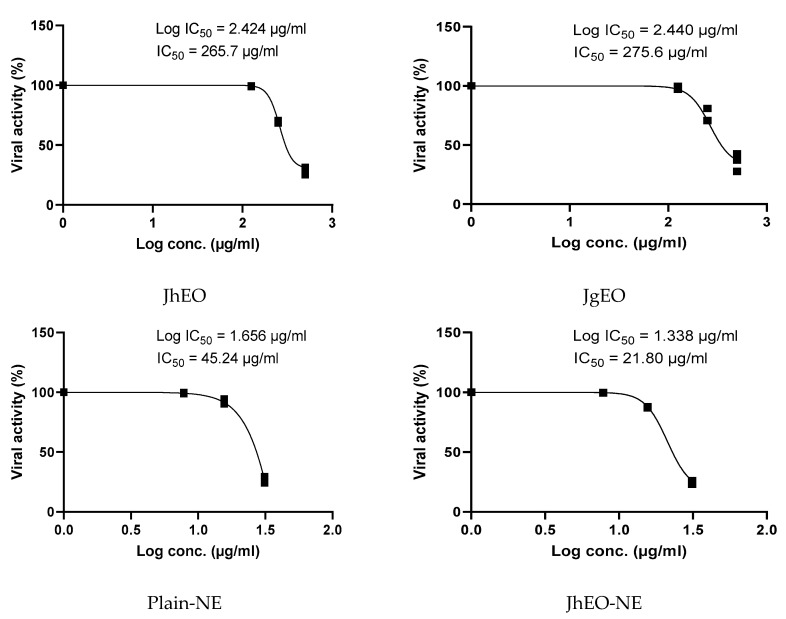
IC_50_ of *J. humile* (JhEO) and *J. grandiflorum* (JgEO) pure essential oils, their nanoemulsion formulations (JhEO-NE) and (JgEO-NE), plain nanoemulsion formulation (Plain-NE) as well as standard acyclovir against HAV virus.

**Figure 6 molecules-27-03639-f006:**
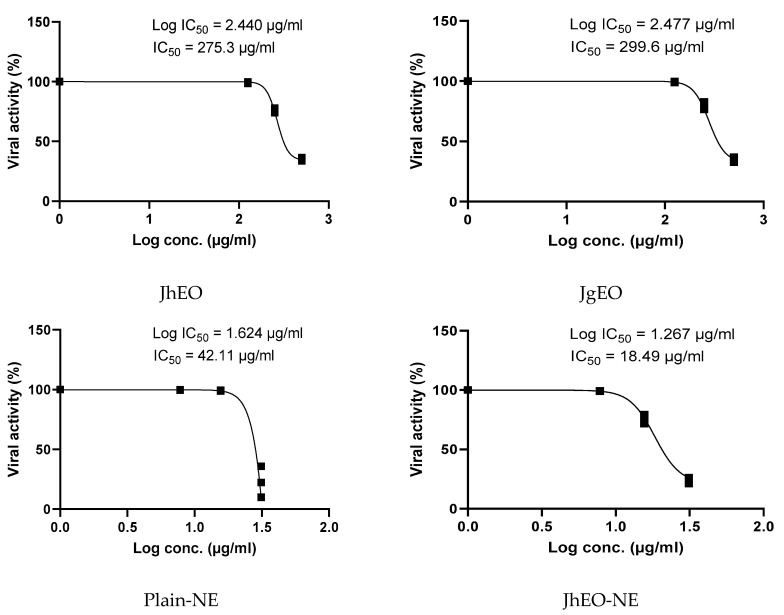
IC_50_ of *J. humile* (JhEO) and *J. grandiflorum* (JgEO) pure essential oils, their nanoemulsion formulations (JhEO-NE) and (JgEO-NE), plain nanoemulsion formulation (Plain-NE) as well as standard acyclovir against HSV-1 virus.

**Table 1 molecules-27-03639-t001:** GC-MS profiling of the essential oils obtained from freshly collected flowers of *Jasminum humile* L. and *Jasminum grandiflorum* L.

	Tentative Identification	R.R.I. calc. ^a^	R.R.I. lit. ^b^	Content %		Base Peak	M. Wt.	M. Formula
				*J. humile*	*J. grandiflorum*			
1	Linalool	1095	1096	17.2	3.6	71.1	154.0	C_10_H_18_O
2	Benzyl acetate	1161	1162	-	32.4	108.0	150.0	C_9_H_10_O_2_
3	Carvone	1240	1243	2.2	-	82.0	150.0	C_10_H_14_O
4	Methyl anthranilate	1338	1337	2.1	2.5	119.1	151.1	C_8_H_9_NO_2_
5	(*Z*)-jasmone	1390	1392	6.6	8.5	79.1	164.0	C_11_H_16_O
6	(*Z*)-caryophyllene	1407	1408	5.6	6.2	41.1	204.0	C_15_H_24_
7	(*E*, *E*)-*α*-farnesene	1506	1505	6.9	7.6	41.1	204.0	C_15_H_24_
8	(*Z*)-nerolidol	1532	1532	5.0	11.9	69.1	222.0	C_15_H_26_O
9	(3*Z*)-hexenyl benzoate	1567	1566	7.4	1.1	105.1	204.0	C_13_H_16_O_2_
10	*n*-hexadecane	1601	1600	-	1.3	57.1	226.0	C_16_H_34_
11	epi-*α*-cadinol	1639	1640	3.0	-	161.1	222.0	C_15_H_26_O
12	epi-*α*-muurolol	1642	1642	2.9	-	43.1	222.0	C_15_H_26_O
13	(*Z*)-methyl jasmonate	1648	1649	-	1.8	83.1	224.0	C_13_H_20_O_3_
14	(*Z*)-methyl epijasmonate	1679	1679	-	1.6	83.1	224.0	C_13_H_20_O_3_
15	(2*E*, 6*Z*)-farnesol	1714	1715	2.5	-	69.1	222.0	C_15_H_26_O
16	Benzyl benzoate	1759	1759	6.9	7.4	105.1	212.0	C_14_H_12_O_2_
17	(2*E*, 6*E*)-farnesyl acetate	1845	1846	2.0	-	69.1	264.0	C_17_H_28_O_2_
18	Benzyl salicylate	1864	1865	2.6	2.5	91.1	228.0	C_14_H_12_O_3_
19	*n*-nonadecane	1899	1900	-	1.2	57.1	268.0	C_19_H_40_
20	Phytol	1941	1943	-	3.5	71.1	296.0	C_20_H_40_O
21	Isophytol	1947	1947	2.4	-	71.1	296.0	C_20_H_40_O
22	Geranyl benzoate	1957	1959	1.7	-	105.1	258.1	C_17_H_22_O_2_
23	Hexadecanoic acid	1960	1960	3.4	1.2	41.1	256.0	C_16_H_32_O_2_
24	Methyl linoleate	2084	2085	3.1	-	67.1	294.1	C_19_H_34_O_2_
25	*n*-heneicosane	2099	2100	-	3.4	57.1	296.0	C_21_H_44_
26	Oleic acid	2141	2142	1.6	-	41.1	282.1	C_18_H_34_O_2_
27	Phytol acetate	2218	2218	1.5	-	43.1	338.1	C_22_H_42_O_2_
28	Tricosane	2300	2300	2.8	-	57.1	324.0	C_23_H_48_
29	Tetracosane	2400	2400	2.2	-	57.1	338.1	C_24_H_50_
30	Hexacosane	2600	2600	4.6	-	57.1	366.1	C_26_H_54_
31	Nonacosane	2900	2900	1.8	-	57.1	408.0	C_29_H_60_

^a^: Relative retention indices obtained experimentally in this study. ^b^: Relative retention indices reported in previous literature [13].

**Table 2 molecules-27-03639-t002:** Cytotoxic activity of the nanoemulsion formulations of the essential oils obtained from *J. humile* (JhEO-NE) and *J. grandiflorum* (JgEO-NE) as well as the reference drug doxorubicin, against HepG2, MCF-7, and THP-1 cell lines, showing CC_50_, IC_50_, and SI.

Sample		CC_50_ ± SD (µg/mL)	Cytotoxic Activity					
			HepG-2		MCF-7		THP-1	
			IC_50_ ± SD (µg/mL)	SI	IC_50_ ± SD (µg/mL)	SI	IC_50_ ± SD (µg/mL)	SI
**Media**	DMSO (-Ve Control)	NT	NA	-	NA	-	NA	-
**Pure EOs**	*J. humile*	362.83 ± 15.31	289.10 ± 5.61	1.26	304.13 ± 3.55	1.19	265.60 ± 10.85	1.37
	*J. grandiflorum*	377.33 ± 10.82	324.90 ± 22.85	1.16	327.53 ± 6.06	1.15	286.37 ± 7.63	1.32
**Nano-emulsion**	*J. humile*	64.28 ± 4.17	22.58 ± 0.86	2.85	36.19 ± 0.77	1.78	51.76 ± 6.68	1.24
	*J. grandiflorum*	62.42 ± 2.07	26.65 ± 0.06	2.34	36.09 ± 1.44	1.73	54.41 ± 0.60	1.15
**Standard**	Doxorubicin	114.00 ± 2.17	33.96 ± 2.03	3.36	52.73 ± 2.44	2.16	47.98 ± 2.35	2.38

NT: Nontoxic, NA: Not active.

**Table 3 molecules-27-03639-t003:** Results of the antiviral assay of the nanoemulsion formulations of the essential oils obtained from *Jasminum humile* L. (JhEO-NE) and *Jasminum grandiflorum* L. (JgEO-NE) as well as the reference drug acyclovir showing CC_50_, MNTC, IC_50_, and selectivity indices (SI) against HAV and HSV-I viruses.

Sample		CC_50_ ± SD (µg/mL)	MNTC (µg/mL)	HAV		HSV-1	
				IC_50_ ± SD (µg/mL)	SI	IC_50_ ± SD (µg/mL)	SI
**Media**	DMSO (-Ve Control)	NT	NT	NA	-	NA	-
**Pure EOs**	*J. humile*	362.83 ± 15.31	500	265.70 ± 4.62	1.36	275.33 ± 7.41	1.32
	*J. grandiflorum*	377.33 ± 10.82	500	275.57 ± 7.51	1.37	299.63 ± 19.76	1.26
**Nano-emulsion**	*J. humile*	64.28 ± 4.17	31.25	21.80 ± 1.17	2.94	18.49 ± 1.20	3.48
	*J. grandiflorum*	62.42 ± 2.07	31.25	25.37 ± 1.26	2.45	31.34 ± 2.72	1.99
**Standard**	Acyclovir	149.97 ± 5.46	31.25	15.04 ± 1.38	9.98	12.49 ± 1.98	12.00

NT: Nontoxic, NA: Not active. Bold value denotes the highest activity.

## Data Availability

Data are available upon request to the corresponding authors.

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
