# Peer review of "Nanoemulsions of Jasminum humile L. and Jasminum grandiflorum L. Essential Oils: An Approach to Enhance Their Cytotoxic and Antiviral Effects"

_molecules, 2022, doi:10.3390/molecules27113639_

Round 1

Reviewer 1 Report

The work received for review is interesting but contains a large number of errors and omissions.

The THP1 lineage cells used in the study are used more to assess the effect on inflammation and not on the tumour process. In the case of haematology, monocytic leukaemias are rare and rather another line should be chosen to assess anti-tumour activity. Therefore, the results and conclusions regarding the THP1 line should be described again.

Please specify the source of the cells, the presented company does not conduct this type of activity and the cell cultures were obtained from another source e.g. ATCC. Please also complete the culture conditions, reagents used, time to thaw, cell volumes and other necessary information.

The graphs presented can be removed (they are nightmarish). In the paper itself, dose-response curves of cytotoxicity should be added and described. In addition, data on applied doses should be added to the methodology.

The conclusions and description of the results are too short. The results obtained should be related to the data concerning the characteristics of the lines used. The conclusions should describe how the nanoemulsions obtained can be used.

Reviewer 2 Report

This manuscript evaluated the cytotoxic and antiviral effects of essential oils of Jasminum humile and Jasminum and grandiflorum. Nanoemulsions were prepared to incorporate the essential oils obtained from the flowers of the selected species. The choice of nanoemulsion formulation by the authors is ideal for such oils. The formulations were characterized, evaluated for stability, and assessed for cytotoxicity and antiviral efficacy. The data presented here signifies the potential of the essential oils in both cytotoxicity and antiviral efficacy. I have a few comments to improve this manuscript.

Comments

  1. It is not clear why the authors choose these two flowers and evaluated them for both cytotoxicity and antiviral properties. The introduction does not provide any information regarding the background of this study.
  2. How do the authors fix the composition of the nanoemulsion? Is there any preliminary studies or ternary phase diagram made to fix the quantity? This information is a must.
  3. The main concern of the article is the inconsistency of the content. Hence required a thorough review. For instance, the plant species should be in italics.
  4. GC-MS profiling of the essential oils shows a wide range of compounds in these species. I am Not sure what the authors feel about the compound which showed the activities evaluated in this study. Literature may provide good information, so a suggestion could be made.
  5. I don’t see the flow cytometric analysis diagram?
  6. Abstract; The abbreviations mentioned are not described. It should be expanded when it appears first. For instance, JgEO-NE, JhEO-NE, etc.

Round 2

Reviewer 1 Report

Dear authors. 

The revised manuscript incorporates much of the comments from the previous review.

I still have the following comments:

1. The authors indicate as source of cells the Tissue Culture Unit, VACSERA, unfortunately I cannot verify this source. I have not found this laboratory in any publication as a source for obtaining cell cultures. I kindly ask you to add the original origin of the lines used. In the description, please leave the lab that gave you the cells. Lack of this information affects the assessment of the quality of the tests performed. Additionally, please complete the description of the culture media used and the conditions under which the cells were cultured before the experiment. This is a standard description in case of performing experiments on cell lines. In case of each of the lines the type and origin of cells should be written in the publication. This is important for the reader, the VERO line can be used in this study but remember it is a monkey line. 

2. Please relate the results obtained to the characteristics and type of lines used in your description.
